# Preparation of NaYF_4_:Tm, Yb, and Gd Luminescent Nanorods/SiO_2_ Nanospheres Composite Thin Film and Its Application in Perovskite Solar Cells

**DOI:** 10.3390/ma16216917

**Published:** 2023-10-27

**Authors:** Qi Luo, Jian Yu, Xueshuang Deng, Ming Cao, Shifang Ma, Qiongxin Hua, Dan Xue, Fenghui An

**Affiliations:** 1College of Materials Science and Engineering, Jiujiang University, Jiujiang 332005, China15697927781@163.com (D.X.); fenghui_an@126.com (F.A.); 2Electronic Information and Electrical College of Engineering, ShangLuo University, Shangluo 726000, China

**Keywords:** perovskite solar cells, spectral conversion, NaYF_4_, antireflection coating, SiO_2_

## Abstract

In this study, we aim to minimize light loss and achieve high power conversion efficiencies (PCE) in perovskite solar cells (PSCs) by employing a spectral conversion film component with antireflection properties. In our scheme, NaYF_4_:Tm, Yb, and Gd luminescent nanorod/silica nanosphere-based thin films are applied on CH_3_NH_3_PbI_3_ PSCs to improve the device efficiency. The film was fabricated by spin coating an aged silica sol containing NaYF_4_:Tm, Yb, and Gd luminescent nanorods. The size and the spectral conversion properties of the NaYF_4_:Tm, Yb, and Gd luminescent nanorods were controlled by tuning the Gd^3+^ ion concentration. The microstructure and the transmittance properties of the thin film were controlled by changing the concentration of NaYF_4_:Tm, Yb, and Gd luminescent nanorod in silica sol. The thin films have excellent spectral conversion properties while exhibiting a maximum transmittance. The photovoltaic performance of PSCs with NaYF_4_:Tm, Yb, and Gd luminescent nanorod/silica nanosphere-based thin films was systematically investigated. The light transmittance was optimized to 95.1% on a cleaned glass substrate, which resulted in an average increase of about 3.0% across the broadband range of 400–800 nm. The optimized films widen the spectrum of light absorbed by conventional PSC cells and reduce reflections across a broad range, enhancing the photovoltaic performance of PSCs. As a result, the PCE of the PSC increased from 14.51% for the reference device without a thin film to 15.67% for the PSC device with an optimized thin film. This study presents a comprehensive solution to the problem of Fresnel reflection and spectral response mismatch of the PSCs, which provides new ideas for the light management of PSCs.

## 1. Introduction

Due to their advantages, such as direct bandgap [1,2,3,4], high carrier mobility [5,6], long electron-hole diffusion length [7,8,9], low non-radiative recombination [10], high internal quantum efficiency [11], low cost [12], and high flexibility [13], perovskite solar cells have great potential to become the next generation of high-efficiency photovoltaic products. Researchers have tried many methods to improve the power conversion efficiency (PCE) of the device, such as synthesis of new materials, bandgap engineering, interface engineering, and optical engineering [14,15,16,17,18,19,20]. After much research and design, the PCE of the PSC has exceeded 26% [21]. The efficiency of PSC devices is affected by several factors, including spectral mismatch and Fresnel reflection loss. PSCs typically have an optical absorption in the wavelength range of 400–800 nm, with the main optical absorption typically in the range of 400–700 nm; their light efficiency is low at 300–400 nm and 700–800 nm, while for the wavelength range < 300 nm or > 800 nm they are almost unusable. Fresnel reflection losses occur when light from the surrounding medium (air) is incident on the absorbing medium of the battery device (usually a high refractive index semiconductor) and are important factors limiting the PCE of the device. Faced with these two limitations, researchers are attempting to convert the near-infrared (NIR) or blue-ultraviolet light in the incident light into the visible range through the up-conversion (UC) or down-conversion (DC) processes so that it can be absorbed and utilized by battery devices or use nanoparticle antireflective films or scattering layers to mitigate these loss mechanisms.

Among the many UC and DC fluorescent materials, lanthanide-doped fluoride nanoparticles have a long fluorescence lifetime and stable optical properties, making them more suitable materials for photovoltaic devices. Among them, the more typical matrix crystals are NaYF_4_, NaGdF_4_, YF_3_, etc., and different light conversion properties can be obtained by adding different rare-earth elements into them [22,23,24,25,26,27,28]. At present, there are many applications that combine UC or DC materials with photovoltaic devices, some of which are applied to the outer surface of photovoltaic devices and some of which are used in various structural layers inside the device, which has played a certain role in improving the performance of photovoltaic devices [29,30,31,32]. For example, the monodisperse β-NaYF_4_: Yb^3+^, Er^3+^ nanofluorescent particles were modified by diblock copolymers to replace the traditional titanium dioxide (TiO_2_) mesoporous material in PSC, and the PCE of the device was increased to 17.8% [33]. β-NaYF_4_: Yb^3+^, Er^3+^ nano-prisms and β-NaYF_4_: Yb^3+^, Tm^3+^@NaYF_4_ core-shell nanoparticles were mixed with dilute TiO_2_ and deposited on PSCs as spectral conversion mesoporous layers [32,34]. In the above study, it is difficult to attribute the increased efficiency of the cell devices to the increase in photocurrent caused by the conversion of NIR or ultraviolet (UV) light to visible light by the photo-conversion material. In our previous work, the application of DC nanocrystals to the outer surface of the device by combining DC nanocrystals with PMMA thin films increased the PCE of the device by 4.5% and demonstrated that this performance improvement was due to the down-conversion performance of the DC nanocrystals [35]. Xu et al. prepared LiYF_4_: Yb^3+^, Er^3+^ nanocrystals and deposited them on the outer surface of photovoltaic devices, which increased the PCE of the device by 7.9% (seven–eight times the incidence of sunlight) and reached 5.72% PCE under 6.2 W cm^−2^ 980 nm laser, confirming the UC effect of the LiYF_4_: Yb^3+^, Er^3+^ photo-conversion layer on photovoltaic devices [36]. However, for most applications of optical conversion layers, there are few reports on further improvement of the optical transfer layer from a transmission perspective.

High transmittance anti-reflective coatings (ARC) are widely used on the outer surface of various photovoltaic devices to reduce the surface reflection loss of incident light over a wide wavelength range [37,38,39,40,41,42]. At present, ARC is mainly prepared by sputtering, etching, the sol–gel method, etc. [43,44,45,46,47,48]. Among them, the sol–gel method is an ideal preparation method with simple process and low cost. SiO_2_ has many advantages, such as low refractive index, high durability, strong environmental resistance, simple preparation process, controllability, and low price, making it an ideal material for the preparation of ARC. In our previous work, the application of optimized SiO_2_ nanosphere-based ARC to the outer surface of the PSC improved the PCE of the device by 6.8% [49].

Nano-fluorescent materials and SiO_2_ nanosphere-based ARCs are nanostructures, both can be prepared as thin films and applied to the outer surface of photovoltaic devices, and it is possible to combine the two. However, there are currently few reports on this type of material.

In this work, a NaYF_4_/SiO_2_ bi-functional film with spectral conversion and antireflection properties is presented. NaYF_4_: Tm, Yb, and Gd (denoted as NYTYG) nanorods with uniform size, good dispersion, and up- and down-conversion functions were successfully synthesized using the hydrothermal method, and the length, diameter, and luminescence performance of NYTYG nanorods were optimized by adjusting the Gd^3+^ concentration. The optimized NYTYG nanorods were combined with SiO_2_ nanospheres ARC, and the performance of the nanorods was optimized by adjusting the concentration of the nanorods and the spectral conversion; anti-reflective bi-functional nanocomposite films with a transmittance increase of 2–3% and spectral conversion functions were obtained. When applied to PSC battery devices, the PCE relatively increased by 8.0% (from 14.51% to 15.67%). This work proposes that the preparation process combines two functions of spectral conversion and antireflection, which provides a new idea for the application of optical conversion materials and ARC in photovoltaic devices, which is conducive to the development of PSC and the development of optical management schemes for various optoelectronic devices.

## 2. Materials and Methods

### 2.1. Synthesis of NYTYG (Y: Tm: Yb: Gd = 79.5 − x: 0.5: 20: x mol%) Nanorods

NYTYG nanorods were synthesized using the hydrothermal method. First, 1.5 g of NaOH was added to 7.5 mL of de-ionized (DI) water and stirred for 10 min, then 25 mL of ethanol and 25 mL of oleic acid were added until a transparent yellowish solution was formed, followed by 10 mL of 0.2 M RECl_3_ aqueous solution and 5 mL of NH_4_F solution with a concentration of 2 M. After stirring continuously for 10 min, the mixture was transferred to a 100 mL reactor, heated at 200 °C for 2 h, cooled, centrifuged 3 times with ethanol and dispersed in cyclohexane.

To investigate the effect of Gd^3+^ concentration on the morphology and optical properties of NYTYG nanorods, the concentrations of GdCl_3_ (x mol%) and YCl_3_ (79.5 − x mol%) (x = 0, 5, 10, 20, 30, 40) were adjusted while the concentrations of RECl_3_ (10 mL, 0.2 M), YbCl_3_ (20 mol%) and TmCl_3_ (0.5 mol%) were kept constant.

### 2.2. Preparation of NYTYG Photoconversion Nanorods/SiO_2_ Nanospheres Composite Films

Tetraethoxysilane (TEOS) and ethanol were purchased from Sigma Aldrich (Darmstadt, Germany) as the precursor and the solvent, respectively. NH_3_•3H_2_O was obtained from Aladdin Industrial Corporation (Darmstadt, Germany) as the catalyst. The hybrid silica sol was prepared by mixing the precursor materials [50].

Initially, 6 mL of DI water, 70 mL of ethanol, 1.5 mL of ammonia, and 3 mL of TEOS were added to the reactor after stirring for 5 min, and a clear solution was formed by stirring evenly. The nano-SiO_2_ anti-reflection membrane precursor was obtained by aging for 24 h (stand at room temperature).

In addition, NYTYG/SiO_2_ composite film precursor was prepared. To prepare the NYTYG photo-conversion nanorod/SiO_2_ nanosphere composite film precursor, the NYTYG photo-conversion nanorod powder (5, 10, 20, 50 mM) was added to the silica sol before aging. For the control experiment, we also synthesized NaYF_4_: Gd (denoted as NYG) nanocrystals (Gd^3+^ ions are used to control the size of the nanorods) without TmCl_3_ and YbCl_3_ under the same conditions, and prepared NYG/SiO_2_ composite film precursors with a concentration of 10 mM.

### 2.3. Device Fabrication

Substrate preparation: Fluorine Doped Tin Oxide (FTO) glass as the substrate of photovoltaic devices. Before use, the FTO glass was etched with Zn powder and HCl with a concentration of 2 M. Ultrasonic cleaning of FTO glass was carried out with glass cleaning agent, high purity water, and anhydrous ethanol for 30 min each time. After drying at 70 °C, the glass was cooled to room temperature for later use.

Preparation of compact TiO_2_ (c-TiO_2_) layer: 200 μL TTIP and 5 mL ethanol were mixed evenly to form a clarifying solution, which was used as the precursor solution of c-TiO_2_ layer. The FTO substrate was irradiated with UV light for 20 min to improve its surface hydrophilicity. Then, the prepared precursor solution of TTIP c-TiO_2_ layer was spun onto the substrate (4500 r/min, 30 s). After sintering at 500 °C for 30 min, TiO_2_ was cooled with the furnace to complete phase transformation and form compact c-TiO_2_ layer.

Preparation of porous layer: add 444 µL anhydrous ethanol to 0.1 g 18NR slurry and stir overnight to prepare TiO_2_ porous layer precursor solution. The porous precursor was rotated on the substrate with dense layer (2000 r/min, 30 s) and sintered for 30 min at 500 °C, then cooled to room temperature in the furnace.

Post treatment: Substrates with dense and porous layers were immersed in TiCl_4_ aqueous solution (6 mM) for 30 min at 70 °C. After that, they were cleaned with deionized water and ethanol successively, sintered for 30 min at 500 °C, and cooled to room temperature in the furnace.

Preparation of perovskite layer and spiro-OMeTAD hole layer: Adding 0.1975 g of ammonium potassium iodide and 0.5785 g of lead iodide to 1 mL DMSO/butanolactone (DMSO: butanolactone = 3:7) solution, mixing evenly, stirring overnight at 60 °C, the perovskite precursor solution was prepared. Spiro-OMeTAD 0.0724 g was taken, and 1 mL chlorobenzene, 28.8 µL 4-tert-Butylpyridine (TBP), and 17.5 µL Li-tsFI solution was added to it and stirred overnight at room temperature to prepare spiro-OMeTAD hole precursor solution. The perovskite active film was prepared by rotating 50 µL perovskite precursor solution (3500 r/min, 50 s) onto the post-treated p-TiO_2_/c-TiO_2_/FTO substrate, and then heat treatment (110 °C, 30 min) was performed to cool it to room temperature. Subsequently, spiro-OMeTAD hole precursor solution (4000 r/min, 30 s) was coated on it. Finally, Ag/Al was vaporized on the spiro-OMeTAD hole transport layer as metal back electrode.

Preparation of NYTYG photo conversion nanorods/SiO_2_ nanospheres composite film precursor was rotated to the back of FTO glass substrate of the device after back-electrode evaporation, and PSC device with NYTYG/SiO_2_ spectral conversion and antireflection and antireflection dual-function film layer was obtained. After completing all the tests, carefully protecting the PSC device while wiping the back of the FTO glass substrate of the PSC device with a cotton swab dipped in dilute hydrochloric acid can remove the prepared film layer without damage and recover the blank PSC device for reuse. Considering that the photovoltaic performance of PSC devices prepared under the same conditions has a certain fluctuation, all photovoltaic performance tests are based on the same blank PSC devices.

### 2.4. Characterization

The morphologies were characterized using a high-resolution field emission scanning electron microscope (FESEM, Hitachi S4800, Tokyo, Japan). The optical spectra of the composite thin film were investigated and characterized using an ultraviolet–visible light (UV–vis) spectrometer (Hitachi, U-3010, Tokyo, Japan). Photoluminescence (PL) spectra were detected using a HORIBA JobinYvon fluoromax-4 fluorescence spectrophotometer conjunction with a 980 nm diode laser (Kyoto, Japan). Photocurrent density–voltage (J-V) measurements were performed using an AM 1.5 solar simulator equipped with a 1000 W xenon lamp (Model No. 91192, Oriel, Newton, MA, USA). The solar simulator was calibrated using a standard silicon cell (Newport, MS, USA). The light intensity was 100 mW·cm^−2^ at the surface of the test cell. A metal aperture mask with an aperture of approximately 0.09 cm^2^ was used to characterize the photovoltaic performance of the device. External quantum efficiency (EQE) measurements (74125, Oriel, Newton, MA, USA) were also performed on these solar cells.

## 3. Results and Discussion

Figure 1 shows the X-ray diffraction (XRD) patterns of the as-synthesized NaYF_4_: Tm 0.5%, Yb 20%, and Gd x% (x = 0, 5, 10, 20, 30, 40) samples. It can be seen from the figure that when Gd^3+^ ions are not doped, a heterogeneous peak corresponding to the (111) crystal plane in α-NaYF_4_ (standard PDF card 06-0342) appears at 28.40°, and the position of other diffraction peaks is very consistent with the β-NaYF_4_ standard PDF card 16-0334. After the addition of Gd^3+^, the diffraction peaks are all consistent with the β-NaYF_4_ standard PDF card and no impurity peaks can be detected in the XRD patterns, indicating that NaYF_4_ is more inclined to transition from α-phase to β-phase after doping with Gd^3+^.

Figure 2 shows the SEM morphology of NYTYG nanocrystals with different Gd^3+^ ion concentrations; it can be seen that the synthesized NYTYG is a uniform submicron or nanoscale rod-like crystal. As shown in Figure 2A, when the Gd^3+^ ion is not doped, there are some small particles on the surface of the nanorod, and after doping with the Gd^3+^ ion, there are no more of these small particles (Figure 2B–F). Combining these results with the XRD patterns shows that these small particles are α-NaYF_4_. The SEM morphology also shows that the doping concentration of Gd^3+^ ions has a significant effect on the size of the synthesized nanorods. To better understand the size variation of NYTYG nanorods with Gd^3+^ ion concentration, we plotted the variation of the nanorod length and diameter with increasing Gd^3+^ doping (Figure 3). As shown in Figure 3, with the increasing doping concentration of Gd^3+^ ions, the average length and diameter of NYTYG nanorods showed a significant decreasing trend, especially when the concentration of Gd^3+^ ions increased from 10% to 20%, the average length of nanorods decreased sharply from about 925 nm to about 350 nm and the diameter decreased from 130 nm to 60 nm. This is because when the smaller radius Y^3+^ (1.159 Å) is replaced in the lattice by the larger radius Gd^3+^ (1.193 Å), its dipole polarization becomes higher, the distortion tendency of the electron cloud becomes larger, and the symmetry of the lattice decreases, making the β-phase structure more inclined to form. In addition, the electron charge density at the crystal surface also increases, slowing the rate of F-diffusion to the surface, resulting in a smaller NaYF_4_ size due to mutual repulsion between the accumulated charges.

Changes in the concentration of rare-earth ions, the structure of the crystals, and their morphology usually lead to changes in the optical properties. Figure 4a shows the PL pattern of NYTYG nanorods with different Gd^3+^ concentrations under excitation by a 600 mW laser at a wavelength of 980 nm. It can be seen from the figure that under the excitation of the 980 nm laser, there are three ultraviolet luminescence peaks of 290 nm, 345 nm, and 360 nm, and two blue luminescence peaks of 450 nm and 475 nm, corresponding to the ^1^I_6_ → ^3^H_6_ (290 nm), ^1^I_6_ → ^3^F_4_ (345 nm), ^1^D_2_ → ^3^H_6_ (360 nm), ^1^D_2_ → ^3^F_4_ (450 nm), and ^1^G_4_ → ^3^H_6_ (475 nm) processes of the Tm^3+^ ion, respectively. In addition, when doped with Gd^3+^, a new UV luminescence peak appears at 310 nm, corresponding to the UC process of ^6^P_7/2_ → ^8^S_7/2_ of Gd^3+^ ions. To better understand the relationship between the up-conversion optical properties of NYTYG nanorods and the doping concentration of Gd^3+^ ions, we plotted the relationship between the luminescence intensity of NYTYG nanorods at different wavelengths and the concentration of Gd^3+^ ions. As shown in Figure 4b, when the concentration of Gd^3+^ is 5%, the five luminescence peaks of Tm^3+^ reach the highest, which is due to the doped Gd^3+^ ion making the crystal phase purer and the crystallinity higher, so the luminescence performance is better. However, as the Gd^3+^ concentration continued to increase, the fluorescence intensity of the five luminescence peaks of Tm^3+^ gradually decreased, whereas the luminescence peak at 310 nm of the ^6^P_7/2_ → ^8^S_7/2_ upconversion process of Gd^3+^ continued to increase with the increase in Gd^3+^ concentration and began to decrease when the Gd^3+^ ion concentration exceeded 30%.

Figure 5 describes schematically possible UC populating for NYTYG NCs at the excitation wavelength of 980 nm. In the Yb^3+^, Tm^3+^, and Gd^3+^ tri-doped system, as analyzed in Ref. [51], Yb^3+^ absorbs 980 nm near-infrared light as a sensitizer and is distributed to the ^3^H_5_, ^3^F_2, 3_, and ^1^G_4_ energy levels of Tm^3+^ by continuous energy transfer. The ^1^D_2_ energy level is populated by the cross-relaxation process of ^3^F_3_ + ^3^H_4_ → ^3^H_6_ + ^1^D_2_. It is then populated from ^1^D_2_ to ^3^P_2_ by another energy transfer process and then rapidly relaxes to the ^1^I_6_ energy level. The fluorescence at 310 nm is closely related to the doping concentration of Gd^3+^ ions. As shown in Figure 5, for Gd^3+^ ions, the luminescence process at 310 nm corresponds to its ^6^P_7/2_ → ^8^S_7/2_ UC process, energy cannot be directly absorbed from 980 nm photons due to the large distance between ^6^P_7/2_ and the ground state, its energy comes from an energy transfer process (^3^P_2_ → ^3^H_6_(Tm^3+^): ^8^S_7/2_ → ^6^I_J_(Gd^3+^)) followed by relaxation from the ^6^I_J_ energy level to the ^6^P_7/2_ energy level. In other words, Tm^3+^ acts as a sensitizer for Gd^3+^. Therefore, as the Gd^3+^ concentration increases, the peak intensity at 310 nm shows an upward trend, and at the same time the Tm^3+^ ion loses some of its energy due to energy transfer, resulting in a decrease in fluorescence intensity related to the Tm^3+^ energy level.

We also investigated the DC fluorescence properties of NYTYG nanorods with different Gd^3+^ ion doping concentrations. As shown in Figure 6, NYTYG exhibited a wide DC emission peak in the range of 400–450 nm under 360 nm incident light, and as the Gd^3+^ doping concentration increased, the light intensity showed an upward trend, reaching the peak light intensity at a concentration of 20% Gd^3+^ ion before decreasing. It is possible that a ^3^H_6_ → ^1^D_2_ transition occurs in the Tm^3+^ ion under the 360 nm laser, followed by a cross-relaxation process of the electrons in the ^1^D_2_ energy level (^1^D_2_ + ^1^D_2_ → ^3^H_4_ + ^3^P_0,1,2_) deploying to the ^3^P_0,1,2_ energy level and the electrons deploying at the ^3^P_0,1,2_ energy level transitioning to the ^3^F_2,3_, ^3^H_4_ energy level, resulting in a wide luminescence peak at 400–450 nm. Although the UC fluorescence performance of the nanorods is best when the Gd^3+^ ion concentration is 5%, the DC fluorescence performance is best when the Gd^3+^ ion concentration is 20% and the diameter of the nanorods is nanoscale (60 nm), which is more compatible with the size of the SiO_2_ nanospheres in the ARC, the SiO_2_ nanospheres in the optimized ARC prepared in our previous work have a particle size of about 75 nm [49]. Therefore, nanorods with a Gd^3+^ ion concentration of 20% were selected for the next step.

Figure 7a shows the transmittance spectra of uncoated clean glass (blank sample), spin-coated pure SiO_2_ film, NYG/SiO_2_ film, or NYTYG/SiO_2_ film with different NYTYG nanorod concentrations. Table 1 shows the average (T_mean_) and maximum (T_max_) transmittance values of these samples in the 400–800 nm wavelength range, the average transmittance of the blank sample is 91.5% in the 400–800 nm wavelength range. To confirm the effect of NYTYG nanorod incorporation on transmittance, glass samples were spin-coated with pure SiO_2_ARC and NYG/SiO_2_ films doped with NYG nanorods and without Tm^3+^ and Yb^3+^. It was observed that when the concentration of NYTYG nanorods was 50 mM, their average transmittance decreased to 91.2%, which was lower than that of the blank samples. When the concentration of NYTYG was low (NYTYG doped concentration ≤ 20 mM), the average transmittance of glass samples spin coated with NYTYG/SiO_2_ film was 93.9–94.5%, which was almost unchanged compared to 94.3% of pure SiO_2_ samples, and the average transmittance was 2–3% higher than that of blank samples, indicating that the incorporation of NYTYG nanorods at low doses would not have a significant negative effect on the antireflective properties of the film. If it could be applied to PSC, its antireflective effect will increase the amount of light (in the 400–800 nm band) into the battery device by about 2–3%.

Based on the transmittance values in Figure 7a, the percentage increase in transmittance (ΔT/T_ref_) of glass samples coated with different films was further calculated using blank samples as a reference, as shown in Figure 7b. It can be seen from the figure that as the doping concentration of the nanorods increases, the peak position of ∆T/T_ref_ also varies. The highest peak position of pure SiO_2_ samples was 363 nm, and when NYTYG concentrations were 5 mM, 10 mM, 20 mM, and 50 mM, the highest ∆T/T_ref_ positions occurred at 430 nm, 440 nm, 544 nm, and 730 nm, respectively. In addition, the highest peak position for NYG/SiO_2_ samples was 442 nm (the concentration of NYG is 10 mM), which was close to 440 nm at the NYTYG concentration of 10 mM, indicating that the peak position is mainly affected by the nanorod concentration and has little relationship with the fluorescence performance. Observation of the peak position shows that the peak position of ∆T/T_ref_ exhibits a red shift with the increasing nanorod doping concentration.

The transmission properties of ARCs are related to their thickness and surface morphology. Figure 8 shows the SEM surface topography of NYTYG/SiO_2_ films with different NYTYG nanorod concentrations. It can be seen from Figure 8A that when the NYTYG concentration is 10 mM, a layer of monodisperse SiO_2_ nanospheres is formed on the glass substrate, the size of the nanospheres is about 75 nm, and the NYTYG nanorods are uniformly distributed in the SiO_2_ nanospheres. When the NYTYG concentration is increased to 20 mM (Figure 8B), the NYTYG nanorods begin to agglomerate, and the SiO_2_ nanospheres in regions without NYTYG nanorods are still arranged in a single layer, while those around the NYTYG nanorods are stacked to form a double or multilayer structure. When the NYTYG concentration is increased to 50 mM (Figure 8C), the agglomeration phenomenon becomes more severe, causing a sharp decrease in the single-layer arrangement area of SiO_2_ nanoparticles, resulting in a sharp increase in the film thickness and the volume fraction of the SiO_2_ nanoparticles. The concentration of NYTYG nanorods affects the thickness of the film, and according to the formula, the value of λ reaches minimum reflectance or maximum transmittance will increase with d, resulting in the red shift phenomenon in Figure 7b, increasing the volume fraction of SiO_2_ nanospheres affects the effective refractive index of the film and therefore the transmittance of the film.

Through the tests of current density–voltage (J-V) and external quantum efficiency (EQE), we found that NYTYG/SiO_2_ films prepared at appropriate NYTYG concentrations can not only improve the photovoltaic performance of the device by reducing the reflectivity, but also enhance the photovoltaic performance by adjusting the wavelength of the incident light to a band more suitable for PSC through spectral conversion. Figure 9a,b show the J-V characteristics and EQE spectra of clean PSC devices without spin coating (blanks), PSC devices spin coated with pure SiO_2_ ARC, NYG/SiO_2_ films, and NYTYG/SiO_2_ films with different NYTYG nanorod concentrations. The obtained parameters are listed in Table 2, including *J_SC_*, *V_OC_*, PCE, and *FF*. As shown in Figure 9b, the average EQE values of spin-coated PSC devices with different films in the 400–750 nm band increase in the following order: NYTYG (50 mM) < blank sample < NYG (10 mM) < SiO_2_ < NYTYG (10 mM) < NYTYG (20 mM) < NYTYG (5 mM). This order is slightly different from the order of the transmittance values, which may be due to the different antireflection properties of different films at different wavelengths, so there are slight differences in the enhancement of EQE after matching with PSC. In fact, the average transmittance of the EQE of NYTYG (5–20 mM), SiO_2_, NYG (10 mM) in the wavelength range of 400–750 nm is little different, concentrated at 80–82%, which is 4.0–5.3% higher than the 76.4% of the blank sample. EQE and transmittance show a consistent trend of improvement over a wide wavelength range, and NYTYG nanorods have no photo-conversion properties in this wavelength range, so the improvement of EQE in this range can be attributed to the antireflection properties of the thin films and the increase in optical path length caused by scattering. In addition, it can be seen from the figure that the EQE values of the samples spin coated with NYTYG/SiO_2_ film have an enhancement peak in the 350–400 nm band and the 950–1100 nm band, while the samples spin coated with SiO_2_ film or NYG/SiO_2_ film do not have this enhancement peak, and the peak height increases with the increase in the concentration of NYTYG nanorods. This fully explains that this enhancement is caused by the spectral conversion function of the NYTYG nanorods. The EQE value indicates the light trapping ability, charge separation ability, and charge collection ability of the battery device, and the spin-coated film is only deposited on the outer surface of the battery device and does not affect its internal structure, so the increase in the EQE value can be attributed to the improvement of the light trapping ability. According to the above analysis, it can be seen that there are two reasons for the improvement of the light trapping ability of NYTYG/SiO_2_ film, which improves the ability of the PSC devices. On the one hand, due to the antireflection and scattering properties of the thin film, the transmittance increases, the light energy entering the battery device increases, and the optical path length increases; on the other hand, due to the spectral conversion properties of the NYTYG nanorods in the film, the wavelength of the incident light can be converted to a band that is more compatible with the PSC device, and the utilization rate of the incident light by the device is improved.

As shown in Figure 9a and Table 2, the *V_OC_* of the blank sample is 1.00 V, the *J_SC_* is 20.29 mA·cm^−2^, the *FF* is 71.69%, and the PCE is 14.51%. The order of *J_SC_* and PCE values for PSC devices with different films is as follows: blank sample < NYTYG (50 mM) < NYG (10 mM) < SiO_2_ < NYTYG (5 mM) < NYTYG (10 mM) < NYTYG (20 mM). The PCE of the device with SiO_2_ film is 14.88%, which is 2.5% higher than that of the blank sample, and the PCE of the device with NYG/SiO_2_ film is very close to that of the device with SiO_2_ film, which is 14.96%. This shows that although the incorporation of nanorods has a certain effect on the structure and thickness of the film, its transmittance does not change significantly (SiO_2_: 94.3%, NYG/SiO_2_: 94.1%), so the effect on the PSC photovoltaic performance is small. Through the PCE change of the device with NYTYG/SiO_2_ film, it can be found that when the NYTYG concentration has little effect on the average transmittance value (5–20 mM), the PCE of the battery device increases with the concentration of NYTYG nanorods, and when the doped concentration of NYTYG nanorods is 20 mM, the PCE of the battery device increases to 15.67%, which is 8.0% higher than that of the blank sample and 5.3% higher than that of the device with SiO_2_ film. This further proves that the spectral conversion effect of NYTYG nanorods plays a certain role in improving the photovoltaic performance of the device.

## 4. Conclusions

We developed a NaYF_4_: Tm, Yb, and Gd nanorod/silica nanosphere-based spectral converting ARC within CH_3_NH_3_PbI_3_ PSC and the PCE was significantly improved NaYF_4_:Tm, Yb, and Gd nanorods with uniform size, good dispersion, and both up-conversion and down-conversion functions were prepared using hydrothermal methods. The nanorods were optimized by adjusting the doping amount of Gd^3+^ ions, and it was found that the concentration of Gd^3+^ ions affects the size and luminescence properties of the nanorods. Then the NaYF_4_: Tm, Yb, and Gd/SiO_2_ composite films were fabricated by spin coating an aged silica sol containing NaYF_4_: Tm, Yb, and Gd nanorods and optimized by adjusting the concentration of NaYF_4_: Tm, Yb, and Gd nanorods to achieve a balance between antireflection performance and spectral conversion function. The optimized ARC coating on the cleaned glass substrate achieved a maximum transmittance of 95.1%. It also improved the average transmittance by about 3.0% in a broadband of 400–800 nm. When applied to PSCs, the reflectance of the device in the 400–800 nm range is effectively reduced and the incident light has also been modulated into a band that matches the photovoltaic device, finally increasing the PCE of the device from 14.51% for the non-ARC reference device to 15.67% for the ARC-optimized PSC device. This preparation method is simple, cheap, highly controllable, and can perform optical management of PSCs from a more comprehensive aspect, which is expected to provide some references for light management technologies of solar cells.

## Figures and Tables

**Figure 1 materials-16-06917-f001:**
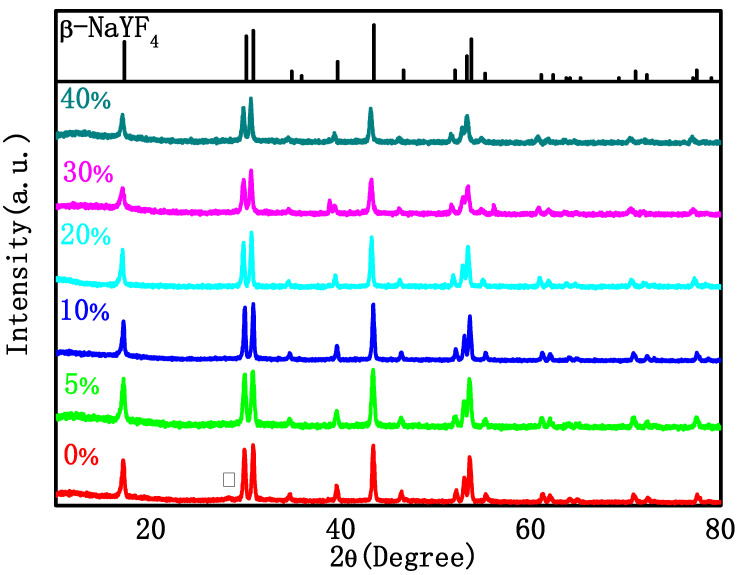
XRD pattern of the NYTYG NCs with different concentrations of Gd^3+^ (From bottom to top: 0%, 5%, 10%, 20%, 40% and β-NaYF_4_ standard PDF card 16-0334).

**Figure 2 materials-16-06917-f002:**
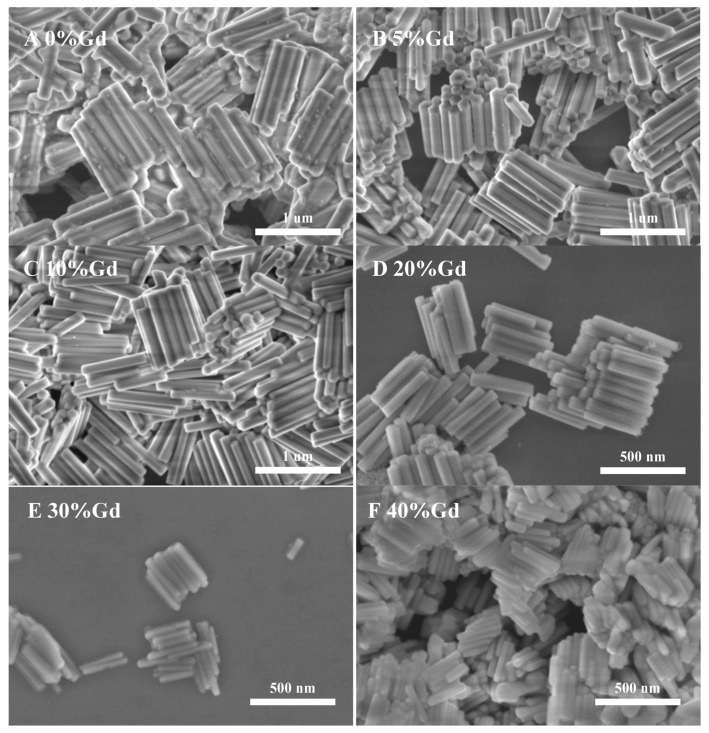
SEM images of NYTYG NCs with different concentrations of Gd^3+^ ((**A**): 0%, (**B**): 5%, (**C**): 10%, (**D**): 20%, (**E**): 30%, (**F**): 40%).

**Figure 3 materials-16-06917-f003:**
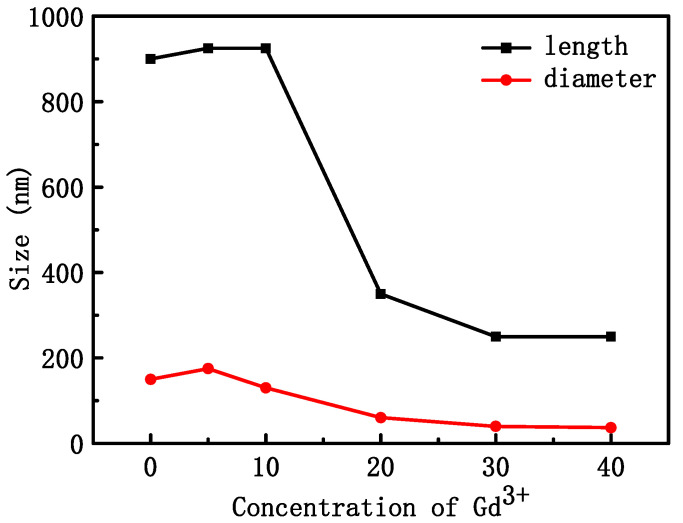
Average length and diameter of NYTYG NCs with different Gd^3+^ concentrations.

**Figure 4 materials-16-06917-f004:**
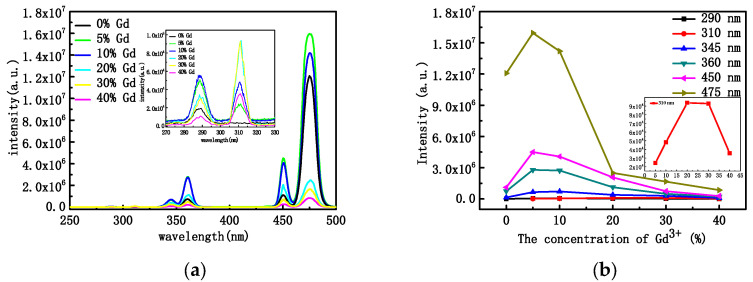
(**a**) PL properties at the excitation wavelength of 980 nm and (**b**) PL intensities of the NYTYG NCs with different concentrations of Gd^3+^.

**Figure 5 materials-16-06917-f005:**
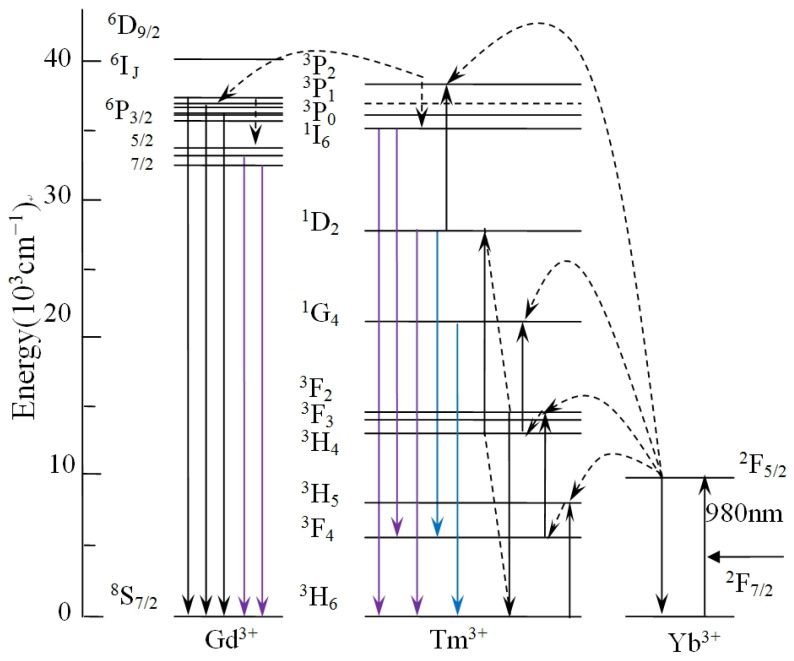
Schematic of UC populating for NYTYG NCs at the excitation wavelength of 980 nm.

**Figure 6 materials-16-06917-f006:**
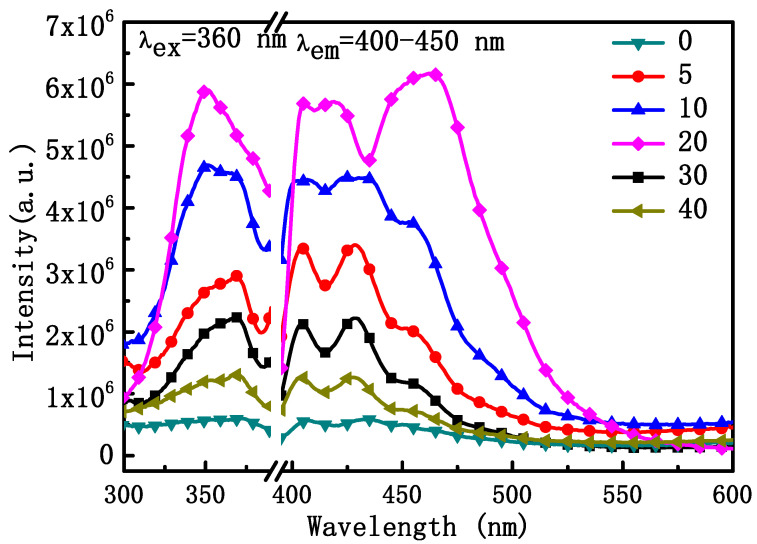
Excitation spectra (**left**) at the emission wavelength of 400 nm and emission spectra (**right**) at the excitation wavelength of 360 nm of the NYTYG NCs.

**Figure 7 materials-16-06917-f007:**
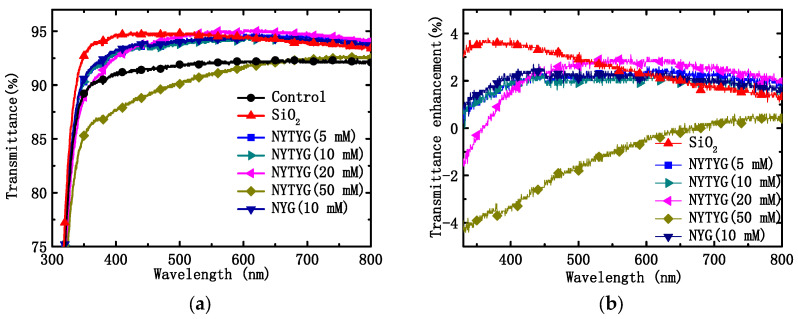
(**a**) Transmittance spectra of the uncoated glass sample and the glass spin-coated with pure SiO_2_ spheres, NYG/SiO_2,_ or NYTYG/SiO_2_ with different concentrations of NYTYG NCs based ARCs and (**b**) transmittance enhancements of different ARCs.

**Figure 8 materials-16-06917-f008:**
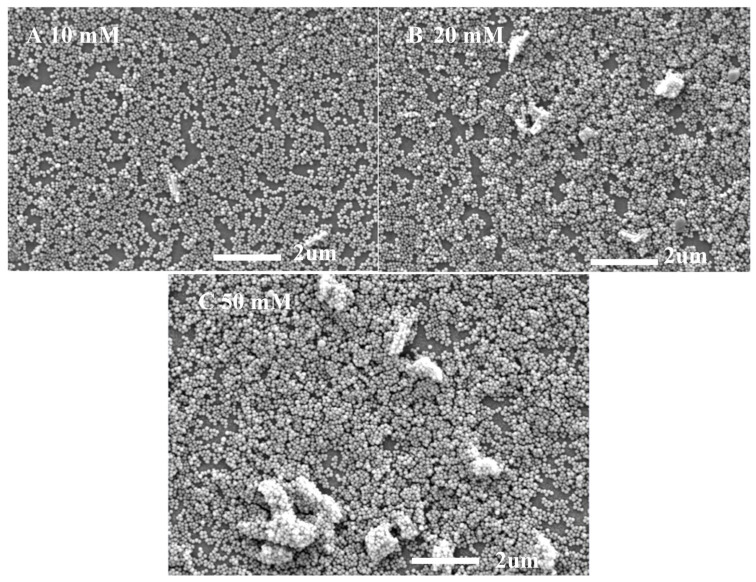
SEM images of NYTYG/SiO_2_ with different concentrations ((**A**): 10 mM, (**B**): 20 mM, (**C**): 50 mM) of NYTYG NC-based ARC grown on glass substrates.

**Figure 9 materials-16-06917-f009:**
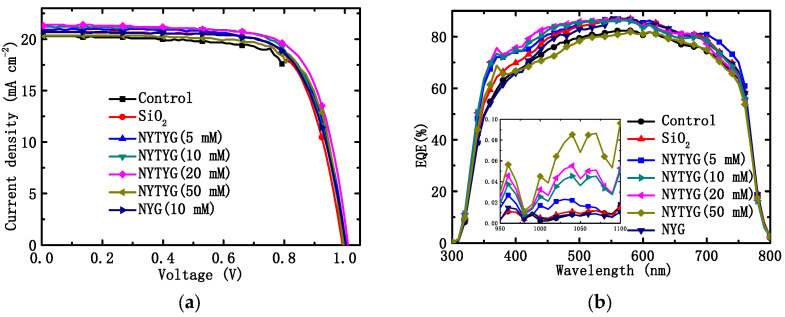
(**a**) *J*-*V* characteristics and (**b**) EQE spectral of PSCs without (control device) and with different ARCs.

**Table 1 materials-16-06917-t001:** Average (T_mean_) and maximum (T_max_) transmittance values of different ARCs.

Sample Concentration	Glass Substrate	NYTYG				NYG 10 mM	SiO_2_
		5 mM	10 mM	20 mM	50 mM		
T_mean_ (%)	91.5	94.1	93.9	94.5	91.2	94.1	94.3
T_max_ (%)	92.4	94.7	94.4	95.1	92.7	94.6	94.9

**Table 2 materials-16-06917-t002:** Parameters of PSCs without (control device) and with different ARCs.

Device	*V*_OC_ (V)	*J*_SC_ (mA/cm^2^)	Fill Factor (%)	Efficiency (%)
Control	1.00	20.29	71.69	14.51
NYTYG (5 mM)	1.00	20.95	71.54	15.02
NYTYG (10 mM)	1.00	21.20	71.82	15.22
NYTYG (20 mM)	1.01	21.34	72.75	15.67
NYTYG (50 mM)	0.99	20.54	72.79	14.74
NYG (10 mM)	1.00	20.69	72.15	14.96
SiO_2_	0.99	20.76	72.26	14.88

## Data Availability

The authors confirm that the data supporting the findings of this study are available within the article.

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
