# Peer review of "Preparation of NaYF4:Tm, Yb, and Gd Luminescent Nanorods/SiO2 Nanospheres Composite Thin Film and Its Application in Perovskite Solar Cells"

_materials, 2023, doi:10.3390/ma16216917_

Round 1
Reviewer 1 Report
The authors submitted a comprehensive study on a composite thin film for enhancement of solar cell efficiency. The composite consists of SiO2 nanospheres and up-converting particles, which properties were studied thoroughly. The research design is correct and the composition of the paper is appropriate. The study is interesting and I recommend it for publication in Materials after a minor revision. There are few minor issues that need to be addressed:
1. The authors could describe the aging process in the Materials and Methods section
2. An optional power dependence study of the luminescence intensity could confirm the processes depicted in Fig. 5.
3. The caption of Fig. 6 is incorrect – the presented spectra are only emission spectra.
4. The authors did not define the meaning of EQE abbreviation.
5. Fig. 7 and 9 have inconsistent color code.
Reviewer 2 Report
The article is well written and structured and both the methods used and the conclusions seem appropriate.
In general, the article is correct and presents interesting results/conclusions. I must say that I found the final analysis of characterization IV very interesting, which completes and justifies the work. However, I must also say that the description of the different luminescence mechanisms, particularly the mechanism that gives rise to the luminescence of Gd, (At this point i this that Down-Conversion (DC) is not the correct word to refer this luminescence)
are not very complete. The mechanisms are well described, but only a process is described without any type of justification or use of references that support what is described. The population of the 6P3/2 level of the Gd is done through 4 Yb-Tm transfer processes (in addition to a cross-relaxation process in the Tm) together with a Tm-Gd energy transfer process. Indeed, this would be one of the possible population processes, but there are no other possibilities? Why have you chosen this one? Is it supported by other experimental data (luminescence life times studies, for example)? Is it supported by previous references? The distance Does interionics play any role in energy transfer? Perhaps this is not the most relevant thing in the general description of the results presented (they should be taken as a minor revision), but knowing the physical processes in detail will undoubtedly help make decisions to try to improve the methods/processes.
In addition, I have detected some connected words (absence of space between words). I suppose the editors will indicate the corrections.
Reviewer 3 Report
In this work, Luo et al. introduce a new solution to improve the conversion efficiency of perovskite solar cells (PSC), aimed at limiting Fresnel reflection and the mismatch between the solar spectrum and the PSC absorption spectrum.
The solution exploits composite thin films incorporating both silicon oxide nanospheres and rare-earth-based nanorods, and demonstrates to be effective in improving transmittance and quantum efficiency as well.
In this sense, the work has a good degree of novelty, and results are quite interesting.
As for the manuscript, it is well-written and organized. State of the art is adequately described in the introductory section. Results are clearly presented and thoroughly discussed. Unfortunately, the same cannot be said about the methodology used, and this is a major issue of the paper, in my opinion, which needs to be properly addressed, as I will better explain later.
In the following, a list of comments and suggestions:
- In the abstract, authors describe as “significant” an improvement of efficiency from 14.51% to 15.67%. However, there is only a difference of about 1% between the two values. I acknowledge that there is an improvement indeed, but I wouldn’t define it as “significant”. I would simply write “enhancing the photovoltaic performance of PSCs”.
- In the introductory section, authors should mention some other important reasons of the current interest in perovskites for photovoltaic solar energy conversion, i.e. their extremely low cost of production if compared to other technologies, and the possibility of being deposited on flexible large-area substrates.
- Line 34. I would rather use “bandgap engineering” than “bandgap management”. The same goes for “interface management” and “optical management”.
- Line 99. Authors write that “the PCE increased by 8.0% (from 14.51% to 15.67%).” However this is a misleading statement, because it increased by about 1% only. It is the initial PCE which increased by 8.0% of its value, and not in absolute terms. Authors should necessarily clarify this point, which may sound tricky.
- In the section describing materials synthesis and preparation, authors write terms like “stirred well”, “several times”, “stirring well”. This not is correct, because it is not a quantitative information. Authors should always mention the duration of stirring, as well as the duration of centrifugation, avoiding to use undetermined quantities.
- As I anticipated before, the major issue of the manuscript is the total lack of description, in the materials and methods section, of the experimental details about the performed measurements. What are the XRD, SEM, and PL experimental details (e.g. XRD source and geometry, and so on). What instruments have been used for electrical current-voltage characterization? What kind of electrical contacts have been used (e.g. contact probes on a workstation, or have the solar cells been equipped with metal deposited electrodes and then mounted on a dedicated PCB)? What experimental setup has been used for spectral EQE measurements (a lamp coupled to monochromator? a lock-in amplifier?) and how has the EQE been calculated from the measured photocurrent values?
- Line 327. Define “EQE” acronym.
- In the concluding remarks, authors define their method as “cheap”. However, the cost of rare-earth materials is rapidly growing nowadays. Do they think it will have a significant impact on the commercial exploitation of their device in the future?
Round 2
Reviewer 3 Report
Revised manuscript is ok.
Authors provided full information on the characterisation methodology as requested.